# The Remaining Mysteries about Brown Adipose Tissues

**DOI:** 10.3390/cells9112449

**Published:** 2020-11-10

**Authors:** Miwako Nishio, Kumiko Saeki

**Affiliations:** 1Department of Laboratory Molecular Genetics of Hematology, Graduate School of Medical and Dental Sciences, Tokyo Medical and Dental University, Tokyo 113-8510, Japan; mnishio.lmg@tmd.ac.jp; 2Department of Regenerative Medicine, Research Institute, National Center for Global Health and Medicine, Tokyo 162-8655, Japan

**Keywords:** brown adipose tissue, BATokine, interscapular BAT, trapezius muscle, cachexia, extracellular vesicles

## Abstract

Brown adipose tissue (BAT), which is a thermogenic fat tissue originally discovered in small hibernating mammals, is believed to exert anti-obesity effects in humans. Although evidence has been accumulating to show the importance of BAT in metabolism regulation, there are a number of unanswered questions. In this review, we show the remaining mysteries about BATs. The distribution of BAT can be visualized by nuclear medicine examinations; however, the precise localization of human BAT is not yet completely understood. For example, studies of ^18^F-fluorodeoxyglucose PET/CT scans have shown that interscapular BAT (iBAT), the largest BAT in mice, exists only in the neonatal period or in early infancy in humans. However, an old anatomical study illustrated the presence of iBAT in adult humans, suggesting that there is a discrepancy between anatomical findings and imaging data. It is also known that BAT secretes various metabolism-improving factors, which are collectively called as BATokines. With small exceptions, however, their main producers are not BAT per se, raising the possibility that there are still more BATokines to be discovered. Although BAT is conceived as a favorable tissue from the standpoint of obesity prevention, it is also involved in the development of unhealthy conditions such as cancer cachexia. In addition, a correlation between browning of mammary gland and progression of breast cancers was shown in a xenotransplantation model. Therefore, the optimal condition should be carefully determined when BAT is considered as a measure the prevention of obesity and improvement of metabolism. Solving BAT mysteries will open a new door for health promotion via advanced understanding of metabolism regulation system.

## 1. Introduction

Brown adipose tissues (BATs), which contribute to non-shivering thermogenesis, has been attracting attention as a target for therapeutic development against obesity and metabolic disorders. BATs are required for the maintenance of body temperature in a cold environment in small mammals, which suffer from large heat loss due to high surface/volume ratios. BATs also contribute to rapid thermogenesis after hibernation. In mice, the presence of BATs has been anatomically determined in specific sites of the body including in interscapular, cervical, axillary, perirenal, and mediastinal regions [1,2] as well as in mammary gland [3]. Humans also have functional BATs [4,5,6,7], whose distribution has been visualized by ^18^F-fluorodeoxyglucose (^18^F-FDG)-PET/CT examinations as fat depots with high glucose uptake capacities. In adult humans, BATs are detected in deep neck regions, supraclavicular regions, axillary regions, paravertebral regions, periaortic regions, and suprarenal regions (Figure 1). The localization of BATs can be summarized as “alongside of large arteries” and “in peri-adrenal gland spaces”, both of which receive abundant adrenergic signals. Since adrenergic stimuli are required for the maintenance of BATs, this distribution pattern of BATs seems highly reasonable.

The amounts of BAT, which tend to reduce with age [8], are inversely correlated with obesity [9,10,11] and glucose intolerance [12]. In humans, interscapular BAT (iBAT) is detectable in newborns and young suckling infants by ^18^F-FDG-PET/CT examinations. There is, however, a difference in the depth of iBAT between mice and humans. In mice, iBAT locates in the subcutaneous space beneath the layer of white adipose tissue (WAT), whereas iBAT locates beneath the trapezius muscle in humans. Although the reason for the disappearance of iBAT in humans during early infancy is unknown, the presence of iBAT in inter-muscular spaces (i.e., between the trapezius muscle and the rhomboid muscles) may be disadvantageous for long-term maintenance of iBAT because skeletal muscles produce substantial heat.

It is known that transplantation of murine BAT [13] or human pluripotent stem cell-derived brown adipocytes (hBA) [14] improves lipid and glucose metabolism [13,14] and prevents obesity [13]. Therefore, approaches to augmenting BAT or preventing its age-dependent reduction may be an effectual strategy for the control of obesity. It is also believed that metabolism-improving effects of BAT is executed via secreted factors, which are collectively termed BATokines. Up to now, several factors have been reported to serve as BATokines including Fibroblast growth factor 21 (Fgf21) [15,16], Interleukin 6 (Il6) [13], Growth differentiation factor (Gdf15) [17], Bone morphogenetic protein 8b (Bmp8b [18], Angiopoietin-like 8 (Angptl8) [19], Neuregulin 4 (Nrg4) [20,21], Slit guidance ligand 2 (Slit2) [22], Ependymin related 1 (Epdr1) [23], and Phospholipid transfer protein (Pltp) [24]. Except for *Angptl8* and *Nrg4*, however, the expression levels of those factors are low in BATs, implying that they serve as autocrine/paracrine factors involved in the regulation of BAT functions rather than hormones that mediate inter-organ communications. Although biological significance of enhanced Angptl8 release by activated BAT remains unclear (reviewed in [25]), Nrg4-Bmp8b axis is suggested to be involved in promoting sympathetic innervation of BAT [26]. On the other hand, hBA secretes a low molecular weight factor (≈800 Da) that enhances insulin secretion by pancreatic beta cells [27]. It is also shown that hBA secretes a variety of extracellular vesicles (EVs) [17], which may serve as BATokines as in the case of WAT [28].

Although BAT is generally accepted as a favorable fat tissue, hyperactivation of BAT may induce unfavorable outcomes. In *Apoe*-deficient mice, hyperactivation of BAT by acute cold exposure promotes atherosclerotic plaque growth and instability [29]. Involvement of BAT in the development of cancer cachexia is suggested (reviewed in [30,31,32,33]). It is even suggested that browning (i.e., acquisition of BAT-like phenotypes) of breast cancer cells and neighboring cells promotes the progression of breast cancers [34], warning that activation of BAT should be kept within an appropriate level when BAT-focused therapeutic development against obesity is considered.

In the following sections, the above-mentioned points are explained in detail for in-depth discussion about remaining mysteries of BATs.

## 2. The Remaining Mysteries about BAT

### 2.1. Localization of BAT

Historically, the presence of murine BATs was anatomically determined while the existence of human BATs was recognized by ^18^F-FDG-PET/CT examinations. Therefore, there was once a controversy as to whether the body distribution of BATs was equivalent between small and large mammals. The findings obtained by nuclear medicine examinations in mice has resolved the dispute. Single photon emission computed tomography (SPECT)/CT scans using a lipid probe, (123/125I)-b-methyl-p-iodophenyl-pentadecanoic acid, along with ^18^F-FDG-PET/CT scans revealed the existence of additional murine BATs that had previously been recognized as unique to humans. Therefore, it is currently accepted that mice and humans share a high degree of topological similarity of BATs [37]. The remaining question is regarding iBAT, which is the largest BAT. While iBAT continues to exist throughout life in mice, it is detectable only in the neonatal period and early infancy in humans by ^18^F-FDG-PET/CT scans. Because distinct results were obtained from SPECT/CT-based imaging and ^18^F-FDG-PET/CT-based imaging of BATs in mice [37], it cannot be easily concluded that adult humans lack iBAT only by the findings of ^18^F-FDG-PET/CT scans.

More than 100 years ago, an anatomical study illustrated the presence of BAT-like tissue in interscapular regions in both newborns and adults in humans [38]. At that time, the presence of BAT in humans was not known and the author called the organ as “interscapular gland”, which show the identical characteristics to iBAT. The author described that it: (1) has a dark brown color; (2) is a paired organ; (3) is lobulated; and (4) is well supplied with blood vessels, all of which correspond to the characteristic of iBAT. The author wrote that “Were it not for the colour and definite outline of the gland, it might be easily mistaken for fat”, suggesting that this organ is not a gland but “a brown colored fat”. Sketches of the histology of this organ further indicated its resemblance with that of BAT. The author also described that “This gland in man is the homologue of the so-called hibernating gland in the rodents”, strongly suggesting that it is iBAT. Interestingly, it was described that this organ showed projections to neck and clavicular regions, indicating that iBAT makes a complex with deep neck BAT and supraclavicular BAT (Figure 2).

Intriguingly, human iBAT is located beneath the trapezius muscle, whereas murine iBAT is located in the subcutaneous space beneath the WAT layer. The difference in the depth of iBAT between mice and humans may come from the difference in the time of migration of the progenitor of the trapezius muscle and that of brown adipocytes during embryogenesis. It is known that the trapezius muscle progenitors migrate from the third and fourth pharyngeal arches to upper back regions [39], while brown adipocyte progenitors arise from a posterior part of somite and migrate to dorsal spaces (reviewed in [40]). If the migration of brown adipocyte progenitors precedes that of the trapezius muscle, iBAT will reach subcutaneous spaces and be located beneath WAT as in the case of mice. On the other hand, if the migration of the trapezius muscle progenitors precedes that of brown adipocyte progenitors, iBAT will be located beneath the trapezius muscle as in the case of humans (Figure 3). Currently, information about iBAT in adult humans is limited. For detailed assessment of human iBAT, nuclear medicine examinations using a new probe are required.

### 2.2. BATokines

Since BAT-depleted mice developed morbid obesity [41] and glucose and lipid metabolic disorders [42], BAT plays indispensable roles in obesity prevention and metabolism improvement. Nevertheless, *ucp1*-deficient mice, which lack BAT-dependent thermogenesis activities, did not undergo obesity under ambient temperature [43]. Therefore, functions other than thermogenesis per se including secreted factor-mediated effects contribute to obesity prevention (reviewed in [8]). Several factors have been reported to serve as BATokines including Fgf21 [15,16], Il6 [13], Gdf15 [17], Bmp8b [18], Angptl8 [19], Nrg4 [20,21], Slit2 [22], Epdr1 [23], and Pltp [24]. Regarding Fgf21, Il6, Bmp8b, Slit2, Epdr1, Pltp, and Gdf15A, BAT is not the major producer as a gene expression database indicates (Figure 4a–d,g–i). It was reported that activated BAT expressed Fgf21 at a level comparable with that of the liver [15]. However, the gene expression database shows that the main source of Fgf21 is not the liver but the pancreas (Figure 4a), which is compatible to the finding of Kuroda et al. [44]. It was even shown that cold exposure or β3-adrenergic stimulation caused a significant induction of *Fgf21* mRNA levels in BAT without a concomitant increase in FGF21 plasma level [16]. Therefore, Fgf21 may serve as an autocrine/paracrine BATokine. Regarding Il6 (Figure 4b) and Gdf15 (Figure 4c), BAT is not the main producer, either. Nevertheless, it was shown that intraperitoneally transplanted iBAT-derived Il6 induced *Il6* expression in the host BATs [13], suggesting that iBAT-derived Il6 serves as a mediator of inter-BAT functional association in the body. Since Gdf15 and Il6 create a mutually gene-inducing cycle [17], it seems that Gdf15 is also involved in this association. Regarding Angptl8 and Nrg4T, BAT is the main producer (Figure 4e,f). However, biological significance of enhanced expression Angptl8 in BAT remains elusive [25]. Moreover, Angptl3/8-deficient mice show upregulated thermogenesis [45], suggesting that Angptl8 may serve as a negative feedback regulator involved in modulating rather than upregulating metabolism. On the other hand, NRG4 is suggested to contribute to the promotion of sympathetic innervation together with Bmp8 [26].

Low molecular weight substances, such as triiodothyronine, retinaldehyde, retinoic acids, free fatty acids, and lactate, also serve as BATokines to contribute to the improvement of metabolism (reviewed in [46]). Although the molecular structure is not yet determined, there is a low molecular weight molecule that enhances basal insulin secretion in the culture supernatant of hBA [27]. Further investigations are required for the discovery of additional BATokines.

### 2.3. Adverse Effects of BAT

Since the rediscovery of human BATs in 2009 [4,5,6,7], they have been expected as an excellent therapeutic target for the treatment of obesity. However, BATs may also exert adverse effects under inappropriately activated conditions.

There are several reports that suggest possible adverse effects of BATs. Under arteriosclerosis-prone conditions with lipid metabolism disorders, hyperactivation of BATs may cause disease exacerbation as reported in *Apoe*-deficient mice, which suffer from atherosclerotic plaque growth and instability when BATs are activated by acute cold exposure [29]. Involvement of BAT in the development of cancer cachexia was also reported (reviewed in [30,31,32,33]). It is known that IL6 (reviewed in [49,50,51]) and GDF15 (reviewed in [52,53,54]) serve as key mediators of cancer cachexia. Interestingly, IL6 and GDF15 are the two “mutually inducing BATokines” that work in an autocrine/paracrine manner [17]. It seems possible that cancer-derived circulating IL6 or GDF15 may trigger the activation of BATs. If the concentrations of plasma IL6 and GDF15 exceed the critical value, positive feedback cycles of inter-BAT mutual activations would be created to induce cancer cachexia as a result of uncontrolled catabolism. BATs may even be involved in the progression of cancer per se. In a xenotransplantation model, where human breast cancer cells were transplanted to mice, BA-selective gene inductions were observed in both transplanted grafts and host microenvironmental cells [34]. It was shown that BA-like phenotype was acquired in early stages of xenografts [34]. Since depletion of BA-tilted cells, which were positive for UCP1 or MYF5, significantly reduced tumor development [34], the acquisition of BA phenotypes is not a result but a cause of cancer progression. Whether this phenomenon is specific to breast cancers or shared among various cancers is not known. There are some affinities between mammary glands and BATs. First, BATs are known to emerge in mammary glands during postnatal development [3]. Secondly, there is a resemblance in the gene expression profile between BATs and mammary glands as they are categorized into the same group in BioGPS [47]. Thirdly, both BATs and mammary fat pads, but not other WATs, are independent of *cebpa* gene in their development. *Cebpa*-deficient mice, which die immediately after birth, are rescued by *cebpa* gene expression in the liver with the albumin promoter. These mice show intact BATs and mammary fat pads, while they lack subcutaneous, epidydimal and perirenal WATs [55]. Although an involvement of BATs in the progression of other cancers remains elusive, risk assessment should be carefully performed when BAT-based anti-obesity therapeutic development is considered.

### 2.4. BAT as a Producer of Extracellular Vesicle (EVs)

Not only soluble factors but also EVs serve as mediators of inter-organ communications. The concept of EVs covers a wide variety of microparticles including secretory autophagosomes [56] as well as exosomes, apoptotic bodies, and oncosomes. It is widely accepted that BAT-derived exosomes contribute to metabolism regulation (reviewed in [57,58]). They can even ameliorate metabolic syndromes in mice with high-fat diet [59]. Fat tissues, especially BATs, are recognized as important sources of circulating miRNAs, which are involved in regulating gene expressions of other tissues including the liver [60]. It was also reported that brown adipocyte-derived exosomal miR-132-3p suppressed hepatic *Srebf1* expression to attenuate lipogenic gene expression [61]. Since WAT-derived EVs are known to contribute to lipase-independent lipid release [28], BAT-derived EVs may also serve as lipid releasers. Although current information about BAT-derived large-sized EVs is limited, hBA is known to secret various kinds of EVs including exosomes and mitochondrial marker-positive large sized EVs [17]. It was recently reported that cell-free functional mitochondria are detected in circulating blood [62]. Since brown adipocytes contain abundant mitochondria, BAT may possibly contribute to systematic mitochondrial supply by secreting functional mitochondria into circulation.

## 3. Discussions and Future Perspective

Ever since the rediscovery of human BATs, very little light has been shed on adult iBAT. Although ^18^F-FDG-PET/CT examinations are incapable of visualizing iBAT in adult humans, its existence was shown by anatomical studies. Interestingly, digital infrared thermal imaging (DITI) of healthy individuals taken from the back side often illustrates that the region covering the trapezium muscle including its tendinous parts is one of the spaces with the hottest dermal temperatures [63]. Because tendinous parts do not produce so much heat compared to muscular parts, the trapezius muscle per se may not be the heat source, but rather, certain tissues existing over or beneath the trapezius muscle would be responsible. Since there are no specific tissues over the trapezius muscle, tissues existing just beneath the trapezius muscle might be the heat producer. It was reported that iBAT of adult humans “extends horizontally from the scapular spine to the center of the interscapular region to form a T-shaped mass over and between the rhomboidii muscles and is entirely covered by the trapezius muscle” [38]. Therefore, iBAT may be responsible, at least in part, for the high thermogenesis in the above-mentioned region. Future studies are required for detailed characterization of iBAT in adult humans.

To obtain insight into the precise distribution of BATs in human body, new approaches are required in addition to the conventional imaging technique such as ^18^F-FDG-PET. We previously reported that intravenously administrated carbon nanotubes coated with poly(2-methacryloyloxyethyl phosphorylcholine-co-n-butyl methacrylate) (PMB-CNTs) provided high-quality images of BATs in mice [64]. We found that PMB-CNTs adhered specifically to the capillary endothelial cells in fat tissues. Therefore, they provide clear images of BATs through near infrared photographing. We also reported that CNTs coated with phospholipid polyethylene glycol (PLPEG-CNTs) provide images of BATs by a different mechanism. We showed that PLPEG-CNTs specifically extravasated from BAT capillaries and the extent of extravasation was augmented by fasting [65], suggesting that CNTs will provide an excellent tool for imaging BATs in accordance with their activities. Regarding CNTs, however, there are concerns in terms of safety for clinical usage. Development of safe probes that extravasate from BAT capillaries in proportion to its activity may provide a feasible technique to image the distribution of human BATs.

Although mice and humans share a high degree of topological similarity of BATs [37], there may be some differences in the mode of the contributions to metabolism regulation. It is known that, in contrast to murine BATs, rat BATs show enhanced lipid accumulations despite upregulated lipid oxidization and thermogenesis under cold environments [66]. This is due to amplified uptake of free fatty acid (FFA), which was produced by WAT as a result of enhanced lipolysis [66]. In addition, cold acclimation did not enhance fat oxidation in rat WATs despite upregulated lipolysis and triacylglycerol resynthesis [66], which indicates that beige adipose tissues may not play significant roles in heat production or energy expenditure under cold environments. Although it remains elusive whether human BATs are analogous to rat BATs or murine BATs, the system for metabolism regulation generally depends on the body size of the species reflecting surface/volume ratios. Since body wights of rats (~800 g) are larger than those of mice (~40 g), rat BATs would provide useful information when considering detailed characteristics of human BATs. In this regard, we observed an interesting phenomenon that b-adrenergic stimuli enhanced lipid accumulations (Figure 5a) in human embryonic stem cell-derived brown adipocytes (hES-BA) despite augmented mitochondrial respiration and thermogenesis [14]. Since the culture medium of hES-BA contained abundant FFA [14], FFA uptake by hES-BA was satisfactorily upregulated by b-adrenergic stimuli as in the case of rat BATs under cold environments (Figure 5b).

Many of the approaches to identify BATokines have been taken on the assumption that BATokine secretions are upregulated under conditions where BAT activities are enhanced. However, this may not always be the case. We observed that the titer of insulin secretion-stimulating molecule that exists in the supernatant of hES-BA [27] was not influenced by b-adrenergic stimuli, suggesting that this molecule is continuously secreted to regulate basal metabolic rate. Interestingly, this molecule augments basal insulin secretion without being affected by glucose concentrations in culture medium [27]. BATokines may contribute to metabolism regulation in various modes of actions, some of which are dependent on BAT activity and others are not.

BATs may be a double-edged sword, exerting favorable effects under obesity-related conditions but providing unbeneficial outcome under cancer-related conditions. To understand the optimal condition of BATs toward health promotion, tools that precisely reflect the activity and the volume of BATs in human body is required. For example, identification of the specific serum markers that indicate the total BAT volumes and active BATs volumes may be useful. To advance our understanding of BATs and apply the correct information of human BATs to clinical purposes, we have to solve at least the following three questions: (1) What animals (e.g., mice and rats) are the best model to understand the involvement of human BATs in metabolism regulation? (2) What are the genuine BATokines that are professionally involved in inter-tissue communications for metabolism regulation? (3) What molecules serves as serum markers of human BATs that provide the quantitative value for the total BAT volume and that for active BAT volume?

There remain a number of mysteries about human BATs, which should be explored toward advanced understanding of BATs to make the best use of them for the improvement of human health.

## 4. Conclusions

Human BAT is basically equivalent to that of mice in terms of body distribution and functions. To obtain more detailed and precise information regarding the distribution of BAT in humans, other imaging techniques than ^18^FDG-PET/CT are required. Although BAT is believed to exert anti-obesity effects via secreted factors, genuine BATokines that are specifically produced by BAT to regulate the functions of distant organs have not yet been identified. Since there is a possibility that BAT promotes cancer progression, safety tests should be carefully performed when BAT is considered as a therapeutic target against obesity.

## Figures and Tables

**Figure 1 cells-09-02449-f001:**
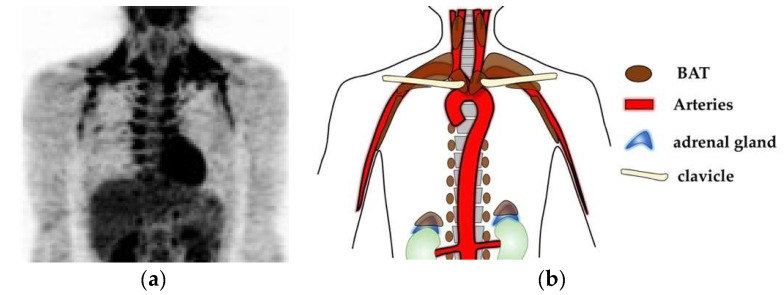
BATs are located alongside of large arteries and in peri-adrenal gland spaces. (**a**) A ^18^FDG-PET/CT examination result of a young adult with acute cold exposure at 19 °C for 2 h. Adopted from Figure 7B in Reference [14].; (**b**) Schematic presentation of body BAT localization in humans. BATs are located in deep neck regions, supraclavicular regions, axillary regions, paravertebral regions, and periaortic regions, which can be summarized as alongside of large arteries, as reported in the case of deep neck BAT [35], and in suprarenal regions, which have been determined as peri-adrenal gland spaces [36]. These regions receive abundant adrenergic signals, which contribute to the maintenance of BAT.

**Figure 2 cells-09-02449-f002:**
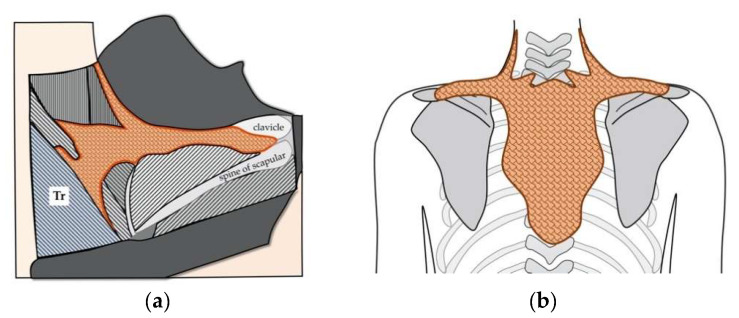
The existence of iBAT in adult humans. (**a**) An anatomical study of humans reported the presence of “a brown colored fat depot” beneath the trapezius muscle (Tr) [38]. It was described that this fat “extends horizontally toward the centre of the interscapular region, where it forms a somewhat T-shaped mass over and between the rhomboidii muscles” and “is entirely covered by the trapezius muscle” [38]. This brown fat depot reportedly possesses morphological and histological characteristics of BATs [38]. Figure was created by referring to Figure 2 in Reference [38]. (**b**) Schematic presentation of iBAT in adult humans drawn according the descriptions in Reference [38].

**Figure 3 cells-09-02449-f003:**
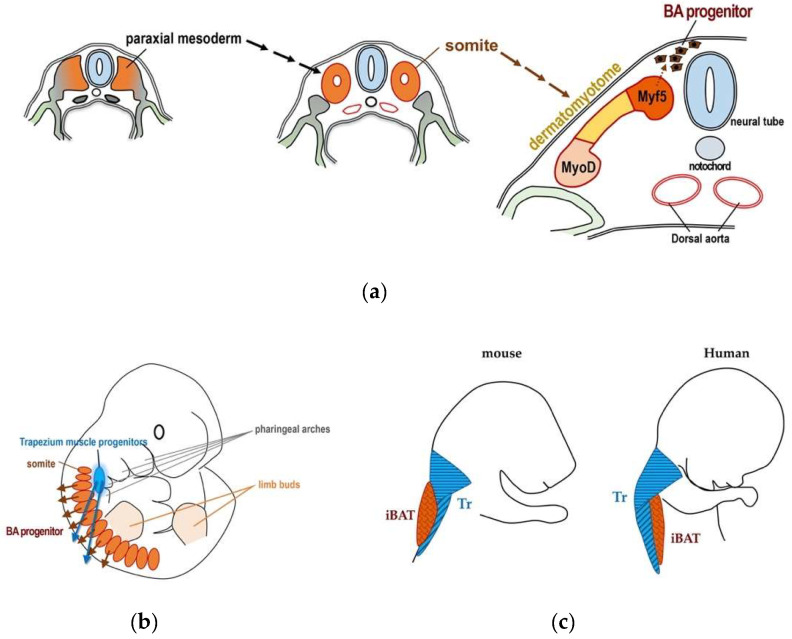
Development of BAT during embryogenesis. (**a**) Schematic presentation of the development of BAT. Brown adipocyte progenitors are differentiated from Myf5-positive myotomes, which are dorsal portions of dermatomyotomes. Myf5-positive cells also consist of back muscle progenitors. Dermatomyotomes are differentiated from somites, which are derived from paraxial mesoderm. Figure was created by referring to Figure 2 in Reference [40]. (**b**) Schematic presentation of the localization and migration of the trapezius muscle progenitors and those of brow adipocytes. Figure was created by referring to Figure 1 in Reference [39]. (**c**) Schematic presentation of relative positional relationship between the trapezius muscle (Tz) and iBAT in mouse (left) and human (right). Figure was created by referring to Figure 1 in Reference [39].

**Figure 4 cells-09-02449-f004:**
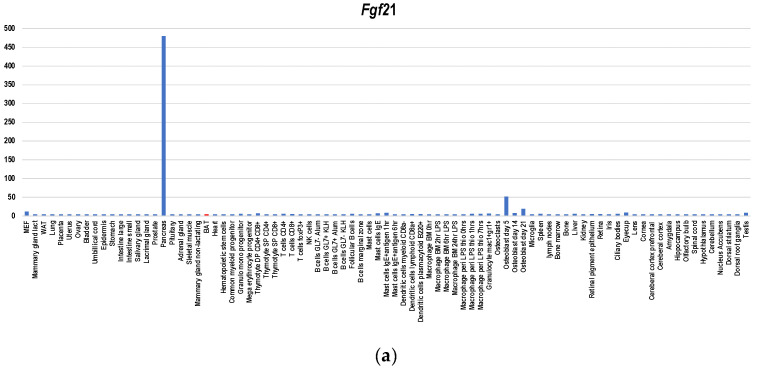
Gene expression profiles of BATokine genes in murine tissues for *Fgf21* (**a**), *Il6* (**b**), *Gdf15* (**c**), *Bmp8b* (**d**), *Angptl8* (**e**), *Ngr4* (**f**), *Slit2* (**g**), *Epdr1* (**h**) and *Pltp* (**i**). The information regarding gene expressions in each BATokine gene was retrieved by searching the data in BioGPS database [47] using a dataset of GeneAtlas MOE430, gcrma [48]. Although this dataset contains not only the gene expression data of murine tissues but also those of cell lines, only the former were used in creating graphs.

**Figure 5 cells-09-02449-f005:**
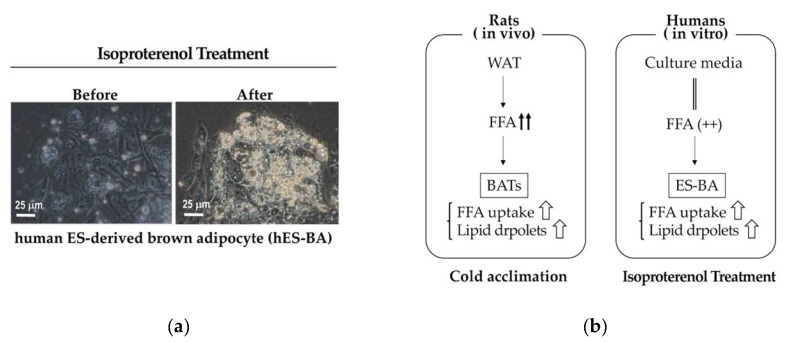
BAT activation enhances the formation of lipid droplets in rats and humans. (**a**) hES-BA [14] was treated with isoproterenol (100 nM) for 4 h and phase contrast micrographs were taken before and after the treatment. (**b**) Schematic presentation the effects of cold acclimation of rats [64] and those of b-adrenergic stimuli of hES-BA. In both cases, lipid accumulation is augmented as a result of upregulated FFA uptake by activated brown adipocytes.

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
