# Peer review of "The Remaining Mysteries about Brown Adipose Tissues"

_cells, 2020, doi:10.3390/cells9112449_

Round 1

Reviewer 1 Report

This is a well written review that will contribute to the literature.

Author Response

We are sincerely grateful for the reviewer's kindly reading our manuscript. It is a great honor for us. Thank you very much.

Reviewer 2 Report

  1. New perspective should be included in the review article.
  2. Full name of BAToktin (ex. Fgf21...) should be spell out before the use of abbreviation. 

Author Response

Thank you very much for giving us excellent comments.
Below, we show point-by-point responses.

The reviewer's comment 1:
New perspective should be included in the review article.
Our response to the comment 1:
According to the kind advice by the reviewer, we added descriptions regarding new perspective for human BAT in section 3, whose subtitle has been changed from "Discussion" to "Discussion and future perspective", in our revised manuscript.

The reviewer's comment 2:
Full name of BATokines (ex. Fgf21...) should be spell out.
Our response to the comment 2:
According to the comments by the reviewer, we wrote the non-abbreviated name of each BATokine in the first appearance.

Reviewer 3 Report

This is a comprehensive and well-written review summarizing the last evidence on brown adipose tissue, its sectred factors and the possible role in the development of some unhealthy conditions.

Some minor suggestions:

  1. Please add the aim of the review in the abstract
  2. Reference number should be added in the reference list
  3. Please add some line on future perspective in the discussion: new possible approaches to identify BAT and promote its activity

Author Response

Thank you very much for giving us excellent comments.
Below, we show point-by-point responses.

The reviewer's comment 1:
Please add the aim of the review in the abstract.
Our response to the comment 1:
According to the comments by the reviewer, we added the aim of the review in the abstract.

The reviewer's comment 2:
Reference number should be added in the reference list
Our response to the comment 2:
We deeply apologize that we had forgotten to delete the default format of the reference list, which appeared as reference 65 ~74 in our previous manuscript. In our revised manuscript, we have deleted it.

The reviewer's comment 3.
Please add some line on future perspective in the discussion: new possible approaches to identify BAT and promote its activity.
Our response to the comment 3:
According to the kind advice by the reviewer, we added descriptions regarding future perspective for human BAT in section 3, whose subtitle has been changed from "Discussion" to "Discussion and future perspective", in our revised manuscript.

Reviewer 4 Report

The primary focus of this review article is not clear; the organization and structure of the paper are confusing.

The mystery behind the browning of WAT, an important aspect of targeting BAT for obesity therapy, is not discussed.

The manuscript started as a review paper but then proceeded as a research article. 

Author Response

Thank you very much for giving us excellent comments.
Below, we show point-by-point responses.

The reviewer's comment 1.
The primary focus of this review article is not clear; the organization and structure of the paper are confusing.
Our response to the comment 1:
According to the kind advice by the reviewer, we revised our manuscript so that the the primary focus of the review article becomes clear and the structure of the paper become well organized.

The reviewer's comment 2. The mystery behind the browning of WAT, an important aspect of targeting BAT for obesity therapy, is not discussed.
Our response to the comment 2:
We appreciate the comment by the reviewer. However, the current review focuses on BAT, but not on beige adipose tissue (i.e. browning of WAT). Since the presence and the physiological significance of beige adipose tissue remain controversial and very difficult theme to discuss, we have concluded that it is not appropriate to include the discussion about the mystery behind the browning of WAT into the current review.

The reviewer's comment 3. The manuscript started as a review paper but then proceeded as a research article.
Our response to the comment 3:
We appreciate the comment by the reviewer; however, we think that, even a review paper, it can send messages to readers, especially to young scientists, to stimulate the researches of the field via presentation an original finding that shows a strong association with a published finding in "Discussion". Therefore, we decided to leave the presentation of our original finding in "Discussions and future perspective" in our revised manuscript.

Round 2

Reviewer 4 Report

BATs should be replaced with BAT throughout the manuscript.